# Acrolein Promotes Aging and Oxidative Stress via the Stress Response Factor DAF-16/FOXO in *Caenorhabditis elegans*

**DOI:** 10.3390/foods11111590

**Published:** 2022-05-28

**Authors:** Jiaqian Hong, Yiming Song, Jiayan Xie, Jianhua Xie, Yi Chen, Ping Li, Danyang Liu, Xiaobo Hu, Qiang Yu

**Affiliations:** State Key Laboratory of Food Science and Technology, China-Canada Joint Laboratory of Food Science and Technology (Nanchang), Key Laboratory of Bioactive Polysaccharides of Jiangxi Province, Nanchang University, Nanchang 330047, China; hongjiaqian777@163.com (J.H.); 13479607955@163.com (Y.S.); xiejiayan_d@163.com (J.X.); xiejianhua7879@163.com (J.X.); chenyi-417@163.com (Y.C.); 17752239585@163.com (P.L.); 15250112296@163.com (D.L.); hxbxq2005@163.com (X.H.)

**Keywords:** *Caenorhabditis elegans*, acrolein toxicity, aging, oxidative stress, DAF-16/FOXO

## Abstract

For this investigation, *Caenorhabditis elegans* (*C. elegans*) served, for the first time, as a model organism to evaluate the toxic effect and possible underlying mechanisms under acrolein (ACR) exposure. The results showed that ACR exposure (12.5–100 μM) shortened the lifespan of *C. elegans*. The reproductive capacity, body length, body width, and locomotive behavior (head thrash) of *C. elegans* were diminished by ACR, especially the doses of 50 and 100 μM. Furthermore, ACR significantly enhanced the endogenous ROS levels of *C. elegans*, inhibited the antioxidant-related enzyme activities, and affected the expression of antioxidant related genes. The increasing oxidative stress level promoted the migration of DAF-16 into the nucleus that was related to the DAF-16/FOXO pathway. It was also confirmed by the significant decrease of the lifespan-shortening trend in the *daf-16* knockout mutant. In conclusion, ACR exposure induced aging and oxidative stress in *C.*
*elegans*, resulting in aging-related decline and defense-related DAF-16/FOXO pathways’ activation.

## 1. Introduction

Acrolein (ACR), a highly reactive α, β-unsaturated aldehyde, has attracted great attention, as it was closely related to food safety and public health [1]. Moreover, ACR is one of the food-processing contaminants that could be widely engendered in thermal processing and industrial discharge, such as the deep-frying of food and the combustion of firewood [2]. Moreover, it can be generated endogenously during lipid peroxidation reactions and the decomposition of polyunsaturated fatty acids, as well as the metabolism of polyamines and amino acids in the body [3]. However, it had been reported that ACR was associated with diverse pathological conditions, comprising lung cancer, cardiovascular disease, and neurological diseases [4,5,6,7]. Studies have found that ACR catabolized endogenous antioxidants and depleted the storage of reduced glutathione in cells, in addition to increasing lipid peroxidation reactions by sparking the reactive oxygen species (ROS) formation [8,9]. The US Environmental Protection Agency has listed ACR as a significant priority poisonous environmental pollutant, while the World Health Organization has established a tolerable oral ACR intake of no more than 7.5 μg/kg bodyweight/day [10]. Though some toxicity endpoints of ACR have been reported, information on the effect on *C. elegans* was not available.

*C. elegans* perches on both terrestrial and aquatic environmental systems throughout the world and has shown to be sensitive to various contaminants. It is not complicated to cultivate worms in the laboratory, since they have both the ability to reproduce in adults in the short term and a short physiological cycle. Moreover, genes in *C. elegans* share a great similarity (60–80%) to human genes, including many homologous genes associated with human disease [11]. As a consequence, *C. elegans* is thought to be an outstanding model animal for research on toxicity, due to its predictability of results in higher eukaryotes (e.g., humans and rats) [12].

ACR has been extensively studied on various cells and organs for the oxidative stress, reproductive toxicity, and aging-related diseases. However, information about ACR toxicity on *C. elegans* is limited, as well as its effect on the oxidative stress and aging process related to the DAF-16/FOXO pathway [13,14,15]. DAF-16 is an essential transcription factor that could integrate different signals from IIS, TOR, AMPK, JNK, and germline signaling pathways to modulate aging via shuttling from the cytoplasm to nucleus in *C. elegans* [16]. Phosphorylated DAF-16 is inactivated by isolation from the cytoplasm, while non-phosphorylated DAF-16 targets the nucleus and controls the expression of genes participating in metabolism, immune defense, and stress resistance. The DAF-16/FOXO nuclear localization can be caused by starvation, heat, and oxidative stress [17,18].

Herein, to further elucidate the toxic effect and the mechanism of ACR on aging-related decline and oxidative stress of *C. elegans*, lifespan, morphology, locomotive behavior, and reproductive capacity of *C. elegans* after ACR exposure would be analyzed. Furthermore, reactive oxygen species (ROS) level, genes, antioxidant enzymes, and DAF-16/FOXO pathway associated to oxidative stress were assessed.

## 2. Materials and Methods

### 2.1. Materials

Acrolein was purchased from Chengdu Aike Chemical Technology Co., Ltd. (Sichuan, China) and stored at 4 °C. The 2′,7′-dichlorodihydrofluorescin diacetate (DCFH-DA) was brought from Sigma-Aldrich (St. Louis, MO, USA). Assay kits for SOD, CAT, and MDA were purchased from Nanjing Jiancheng (Nanjing, China). Assay kits for protein and RNA extraction were purchased from Beyotime (Shanghai, China). FastQuant RT Kit (With gDNase) and SuperReal PreMix Plus (SYBR Green) were purchased from TIANGEN Biotech Co., Ltd. (Beijing, China). All other reagents used were of analytical grade.

### 2.2. C. elegans Maintenance and Bacterial Strains

The *C. elegans* strains used in this work were N2 (wild type), CF1038 (*daf-16*(mu86) I), and TJ356 (zIs356 [*daf-16p*::*daf-16a/b*::GFP + *rol-6*(su1006)]), which were kept in an incubator at 20 °C in Petri dishes containing NGM (nematode growth media) within *Escherichia coli* OP50 (*E.coli* OP50). Age-synchronized *C. elegans* were obtained by treating the adults with a lysis solution (5 M NaOH and 5% NaClO) for 5 min. *C. elegans* at synchronous-stage L1 juvenile were selected for ACR treatment.

### 2.3. Lifespan Assay

The nematodes (N2, CF1038) were treated with different concentrations of ACR (12.5, 25, 50, and 100 μM) on the NGM plate for 48 h. During the reproductive period, *C. elegans* were transferred daily to freshly prepared plates to separate the adults from their progeny.

### 2.4. Development Assay

To better evaluate the developmental condition, analyses of body length and body width were preformed, and *C. elegans* were exposed to different concentrations of ACR, as previously described. Approximately 30 worms were measured by mounting on a 2% agarose pad containing a few drops of 30 mM NaN_3_ for anesthetic fixation on microscopic slides.

### 2.5. Reproduction Assay

To score the number of eggs in the uterus, worms were individually transferred to a drop of bleaching solution on a glass slide covered with an agar pad. The bleaching solution dissolved the body of *C. elegans*, and the eggs were scored immediately under a stereomicroscope.

### 2.6. Locomotion Behavior Assay

*C. elegans* head thrash frequency is a manifestation of athletic ability. *C. elegans* were exposed to different concentrations of ACR, as previously described, followed by being adapted to the NGM plate without *E. coli* OP50 for 1 min. The frequency of head thrashes of the *C. elegans* was recorded within 30 s. Thrashing the head of *C. elegans* from side to side and then back was counted as 1 head thrash.

### 2.7. Accumulation of Reactive Oxygen Species (ROS)

*C. elegans* were prepared as mentioned before and then washed out from the media treatment, followed by exposure to 0.5 mM of H2DCF-DA (2,7-dichlorofluorescein-diacetate) for 1 h at 35 °C. The ROS level was measured immediately after the incorporation of the reagent by a microplate reader (Ex: 485 nm; Em: 538 nm) (Varioskan Flash, Thermo, Waltham, MA, USA).

### 2.8. Antioxidant Enzyme Activity

*C. elegans* (approximately 3000) were treated with ACR for 48 h, as described above, and repeatedly washed 3~6 times with M9 buffer to remove all *E. coli* OP50. Then the supernatant of the *C. elegans* was obtained after homogenizing in M9 buffer and centrifugating (10,000 rpm/min, 10 min, 4 °C). Then the activity of superoxide dismutase and catalase (SOD and CAT) was detected by using corresponding kits (Nanjing Jiancheng Bioengineering Institute, Nanjing, China), according to the manufacturer′s instructions, as well as the malondialdehyde content. The units of enzymatic activity (U) refer to the amount of enzyme that can convert 1 μmol of substrate in 1 min. Finally, the optical absorbance was read by a Microplate Reader. Three independent experiments were performed.

### 2.9. Quantitative Real-Time Reverse-Transcription Polymerase Chain Reaction (qRT-PCR) Analysis

The total RNA of washed *C. elegans* (approximately 3000) was extracted through RNAeasy™ Plus Animal RNA Isolation Kit, and cDNA was synthesized by a FastQuant RT Kit (with gDNase). Moreover, qRT-PCR was performed with SuperReal PreMix Plus (SYBR Green), using the QuantStudio™ 7 Flex Real-Time PCR System (Applied Biosystems, Foster City, CA, USA). The expression levels were normalized to the housekeeping gene β-actin with the comparative 2^−ΔΔCt^ (Ct, cycle threshold) method. The designed primer sequences are listed in Appendix A.

### 2.10. Statistical Analysis

Analyses and statistical significance (considered significant at *p* < 0.05) were performed by using GraphPad Prism version 7 (GraphPad Software, San Diego, CA, USA) and SPSS software (version 21.0). All experiments were repeated at least three independent times, and the results were presented as the mean value ± SD (standard deviation). The values expressed in percentage (%) were normalized by taking a value of 100% for the control.

## 3. Results and Discussion

### 3.1. Effect of ACR Exposure on Lifespan

Acrolein (ACR), a common hazard in cooking or frying food, had been reported to have dose-dependent toxicity on cell apoptosis of Caco-2, GES-1, and hCSFs [3,19]. However, there were few reports on the toxicity of ACR to organisms and its mechanism. To understand the effect caused by a single dose of ACR to the *C. elegans*, the survival curve, median lethal time, and total lifespan of wild-type animals were analyzed (Figure 1 and Table 1). A leftward shift in the survival curve of the ACR-treated group was observed as compared to the control (0 mM), indicating that the addition of ACR was capable of shortening the lifespan of *C. elegans*. Moreover, the median lethal time was associated with the long-term effect of toxicant exposure, and Figure 1B indicates that the median lethal time of *C. elegans* was decreased significantly with the increasing ACR concentration (*p* < 0.05). In terms of average lifespan, the control group was 16.53 ± 0.94 d, and it was reduced by 10.28%, 13.37%, 19.42%, and 39.69% at different concentrations of ACR, respectively. The lifespan is a quantitative indicator of aging [20]. Previous research studies have found that the intracellular homeostasis system of the organism declined gradually in the process of aging, and this might lead to some changes of physiological indexes, such as oxidative stress levels and exercise capacities [21,22]. As a consequence, it could be inferred that the shortened lifespan of *C. elegans* after ACR exposure might lead to oxidative stress and the change of other physiological indexes related to aging.

### 3.2. Effect of ACR Exposure on Development Indicators and Locomotion Behavior

Given that ACR exposure accelerated the aging process of *C. elegans*, in order to explore how the ACR effect on the locomotion behavior and morphology of *C. elegans*, the head swing frequency, body length, and body width of *C. elegans* were measured. According to the observation and calculation through the use of fluorescence inverted microscope, the body length and width were slightly affected at 12.5 and 25 μM concentrations exposure (Figure 2). However, worms under exposure at 50 and 100 μM ACR showed significant diminution in body length (*p* < 0.01) and width (*p* < 0.01). Morphological indexes, including body length and width, were important indexes to measure the development of worms; it turned out that low-dose ACR had little adverse effect on the growth and development of *C. elegans* within 48 h exposure, but when the ACR concentration exceeded at 50 μM, the growth of *C. elegans* was delayed, and when it came to 100 μM, the growth retardation was more pronounced.

In addition, head thrash frequency, one of the most basic behavioral manifestations of *C. elegans*, was closely related to the aging behavior [23]. Previous studies had confirmed that aging was accompanied by degradation of muscles, resulting in a reduction in the frequency of head thrash [24]. As presented at Figure 3, the head thrash frequency of worms decreased with the increasing ACR exposure concentrations, revealing that ACR had a significant inhibitory effect on the motility of the *C. elegans*, especially at a high concentration (100 μM) (*p* < 0.01). Previous studies found that acrylamide exposure could induce a locomotor defect in *C.*
*elegans* [25]. Moreover, the decreasing trend of the head thrash frequency was similar to that of the average lifespan of worms, as a consequence, and the damage of physiological function might be related to the course of shortening lifespan, thus supporting the hypothesis that ACR exposure might lead to physiological function damage in *C. elegans*.

### 3.3. Effect of ACR Exposure on Reproduction

To investigate the effect of ACR on the reproduction of *C. elegans*, the number of eggs in the worm uterus exposed to various concentrations of ACR was calculated. There was a decreasing trend in the egg number in the worm uterus with ACR treatments in a dose-dependent manner when compared to the control (Figure 4). Consistent with previous conclusions on life shortening and physiological function impairment, the egg number in the uterus of *C. elegans* decreased significantly at the ACR exposure concentrations of 50 and 100 μM, indicating that ACR not only affected the lifespan and physiological behavior of *C. elegans* but also affected the reproduction of *C. elegans*.

The egg number in the uterus has been recognized as a direct indicator of population homeostasis, while reproductive capacity is an important indicator for understanding the aging process [26,27]. Similar reproductive suppression results were found in other species; the oral ingestion of 2 mg/kg/day acrolein disrupted male reproductive system in prepubertal male rats by causing histopathological changes and sperm abnormality [28]. Furthermore, previous findings indicated that cyclophosphamide and acrolein deteriorated oocyte quality by altering microtubule spindle structure and chromosomal alignment [29]. For the mechanism of ACR-induced reproductive toxicity, Jeelani et al. found that excessive production of ROS led to mitochondrial damage in mice after ACR exposure, and this might be the main reason for poor oocyte quality and poor reproduction, resulting in poor embryonic development [30]. Consequently, combined with the results mentioned above, it could be inferred that the reproductive toxicity caused by ACR exposure probably is attributed to the elevated levels of oxidative stress.

### 3.4. Effect of ACR Exposure on ROS

ROS is one of the most important parameters in toxicity research, and an excessive increase of ROS in *C. elegans* is able to induce oxidative stress [31]. To study the effect of ACR exposure on the oxidative stress level, the production of ROS in worms after ACR exposure was quantitated by measuring DCF fluorescence. Exposure to ACR resulted in the increased production of ROS in a dose-dependent manner (Figure 5). Furthermore, after exposure to 50 and 100 μM ACR, the ROS levels of worms increased by 35.61% (*p* < 0.01) and 45.74% (*p* < 0.01). This finding was consistent with the observed physiological changes in ACR after exposure in cells; that is, ACR decreased the PC12 cell viability in a ROS-dependent manner [9]. Similar results also appeared in our previous study; the content of ROS in macrophages was significantly increased under the action of acrolein [32]. Moreover, the increasing trend of the ROS level was consistent with the shortening of the lifespan and the variation of the number of eggs in the uterus, suggesting that the previously observed developmental abnormalities and aging in wild-type *C. elegans* might be due to oxidative-stress damage. Meanwhile, the increasing ROS level in worms supported the previous hypothesis that ACR could result in reproductive toxicity through oxidative-stress injury.

### 3.5. Effect of ACR Exposure on Antioxidant Enzymes Activities and Malondialdehyde (MDA) Level

To investigate the effect of ACR on the antioxidant defense system in *C. elegans*, the activities of antioxidant enzymes and MDA content of wild-type worms treated with ACR were detected (Table 2 and Figure 6). The CAT enzyme could be synthesized naturally in the cytoplasm, and after the treatment with a high concentration of ACR, CAT enzyme activity decreased from 16.31 ± 0.08 U/mg prot (0 µM) to 9.94 ± 0.09 U/mg prot (100 µM) and decreased by 39.06%. Moreover, a significant decrease was also observed in the activity of the SOD enzyme, which was known to be a naturally existing superoxide radical scavenging enzyme in organisms. The activity of the SOD enzyme was about 206.20 ± 1.56 U/mg prot in the control group; however, it was significantly reduced by 59.67% under high-dose exposure of ACR (100 µM) (*p* < 0.01), suggesting that the antioxidant defense capacities of *C. elegans* were significantly downregulated after ACR exposure. Furthermore, the MDA levels of ACR-treated *C. elegans* were significantly increased by 44.89% (*p* < 0.01) and 64.21% (*p* < 0.01) in the groups with 50 and 100 µM ACR exposure, respectively. It was worth noting that the content of MDA increased with the increasing concentration of ACR, as was consistent with the change trend of ROS level, indicating that the production of MDA might be associated with the oxidative stress induced by excessive increase in ROS.

Under normal physiological conditions in worms, SOD and CAT enzymes played a crucial role in the elimination of O^2−^ and H_2_O_2_ and kept a dynamic balance with ROS and MDA levels [33]. Our previous study also found that the oxidation/antioxidative balance in macrophage was destroyed under the action of acrolein [32]. In addition, it was reported that ACR-treated rats had significantly elevated MDA levels in their vestibulocochlear nerve tissue [34]. However, after exposure to ACR, the ROS production was excessive and then followed by the accumulation of MDA content, and as a consequence, the balance between SOD, CAT, MDA, and ROS was destroyed, leading to oxidation disorder, which further affected the body health that was embodied in the shortening of the lifespan and the behavioral retardation.

### 3.6. Effect of ACR Exposure on daf-16 Genetic Strains

DAF-16 in *C. elegans* is an orthologue to mammalian FOXO transcription factor and is a major transcriptional regulator related to lifespan; it integrates upstream pathway signals and triggers transcription changes in genes involved in aging, development, stress, and immunity [35,36]. Previous research studies have reported that the proteins encoded by some DAF-16 target genes could protect cells from oxidative stress [37]. According to the results of antioxidant enzymes’ activities mentioned above, ACR exposure enhanced the oxidative stress of worms. Thus, to detect whether ACR exposure activated the transcriptional activity of the DAF-16/FOXO signaling pathway, the TJ356 stain was used to analyze the DAF-16 nuclear translocation. The results showed that 12.5 and 25 μM ACR exposure decreased the cytosolic localization of DAF-16 (decreased to 22.2% and 7.8%), suggesting that the low concentration of ACR activated the antioxidant resistance caused by DAF-16/FOXO (Figure 7). It was consistent with the findings that exposure to pharmaceuticals which induce oxidative stress, such as H_2_O_2_, *N*,*N*′ bis-(2-mercaptoethyl) isophthalamide, juglone, and so on, could lead to the DAF-16 nuclear translocation [38,39,40]. It is worth noting that, with high exposure to ACR, the percentages of DAF-16 nuclear localization were decreased significantly (50 μM, 43.8%; 100 μM, 16.0%) (*p* < 0.01) as compared to those treated with 25 μM of ACR, and this is similar to what was found in chronic exposure of zearalenone [41]. The previous experimental results of this paper had found that, when exposing to 12.5 μM of ACR, the locomotion behavior, e.g., head thrashes, was slightly affected, indicating that low dose exposure hardly affected the normal physiological functions of worms, but activated their own immune defense system. Whereafter, high-dose exposure to ACR might be able to destroy the activated immune defense previously, which was consistent with the significantly increasing ROS level that was detected previously. Therefore, in order to further verify the effect of oxidative stress caused by ACR exposure on DAF-16/FOXO, a lifespan experiment of *daf-16* knockout mutants was conducted. It was found that the lifespan of the mutant (Figure 8) was not significantly shortened as compared to the N2 strain (Figure 1A). Remarkably, the effect of ACR on the lifespan of DAF-16 knockout mutants did not disappear totally. This might be owed to the fact that other transcription factors could participate simultaneously; this idea needs to be investigated in more depth in the future. Previous studies had mentioned that stressful conditions, such as oxidative stress or failure to transmit the signal to DAF-16, could cause internal pressure blocking IIS pathways and then lead to the transformation of DAF-16 from the cytoplasm into the nucleus and induce the transcriptional activity of DAF-16/FOXO [42]. A similar result was found in our research that ACR induced oxidative stress and targeted the DAF-16/FOXO pathway.

### 3.7. Effect of ACR Exposure on Expression of daf-16, ctl-1, and ctl-2

The underlying mechanism of ACR affecting the oxidative stress and lifespan of worms is still unclear at the genetic level. It was known that both of *ctl-1* (encodes an unusual cytosolic catalase) and *ctl-2* (encodes a peroxisomal catalase) were target genes of DAF-16. As shown above, 100 µM ACR exerted the strongest inhibitory effect on the antioxidant enzyme activities and the greatest destruction in the DAF-16 pathway among all concentrations (Figure 6 and Figure 7). Therefore, to investigate the effect on the expression of downstream genes after the possible damage, the same concentration was used for qRT-PCR experiment, and the results were listed in Figure 9. After 100 µM ACR exposure, the expression levels of *daf-16*, *ctl-1*, and *ctl-2* were significantly downregulated by 25.5%, 15.3%, and 28.3%, respectively (*p* < 0.05). The downregulation of these gene expressions was similar to the previous study that chronic exposure to the high concentration of zearalenone increased the ROS level and decreased the proportion of DAF-16::GFP in the nucleus, while decreasing the expression levels of *ctl-1* and *ctl-2* genes [41]. In summary, it could be inferred that excessive ROS caused by excessive exposure to toxins disrupted antioxidant defense systems and further promoted the production of ROS, as well as resulted in aging acceleration. Moreover, the downregulation of genes encoding antioxidant enzymes was in accord with previous findings of a significant reduction in CAT activity, indicating that decreased expression of antioxidant genes would affect the secretion of antioxidant enzymes, which might be related to the increasing ROS level. Altogether, ACR was capable of affecting the expression of related genes downstream of DAF-16, validating the previous inference about the DAF-16 pathway.

## 4. Conclusions

To summarize, this study demonstrated that ACR exposure accelerated the aging of *C. elegans* and had an adverse effect on the reproduction, development, and exercise behavior, which is accompanied by an increasing level of ROS. The dynamic balance between ROS, CAT, SOD, and MDA was broken by the excessive ROS level, leading to activation of oxidative-stress-related pathways. The nuclear localization of DAF-16 could be observed significantly after exposure to ACR and the effect of ACR on lifespan in *daf-16* knockout mutants, indicating that oxidative stress and aging caused by ACR might be associated with DAF-16/FOXO pathway. This research provided the theoretical basis for the toxic effect of ACR and its potential mechanism of oxidative stress in regulating aging, which contributed to enrich the toxicological evaluation of ACR.

## Figures and Tables

**Figure 1 foods-11-01590-f001:**
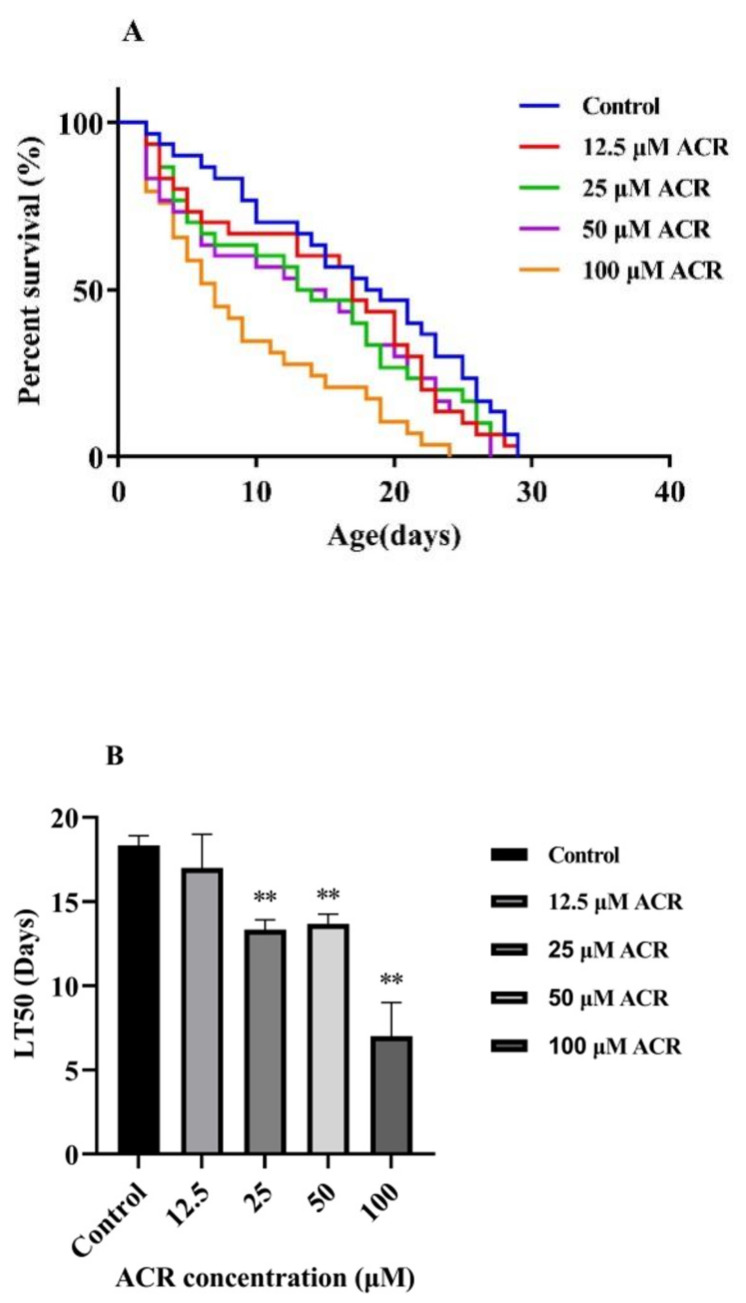
Effect of ACR exposure (12.5, 25, 50, and 100 µM for 48 h) on lifespan (**A**) and median lethal time (**B**) of N2 nematodes. Values are means ± SD. ** Compared with the control group *p* < 0.01.

**Figure 2 foods-11-01590-f002:**
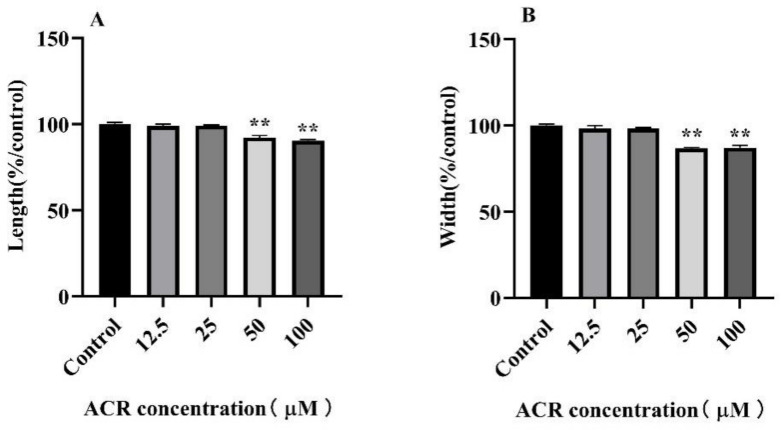
Effect of ACR exposure (12.5, 25, 50, and 100 µM for 48 h) on length (**A**) and width (**B**) of *C. elegans*. Values are means ± SD. ** Compared with the control group *p* < 0.01.

**Figure 3 foods-11-01590-f003:**
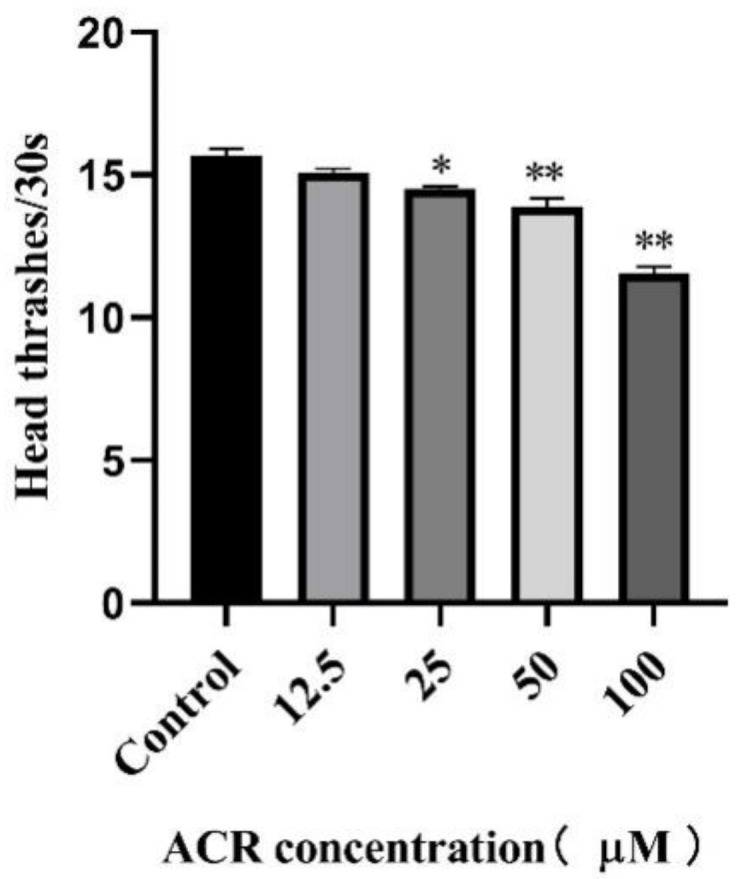
Effect of ACR exposure (12.5, 25, 50, and 100 µM for 48 h) on head thrash frequency of *C. elegans*. Values are means ± SD. * Compared with the control group, *p* < 0.05; ** compared with the control group, *p* < 0.01.

**Figure 4 foods-11-01590-f004:**
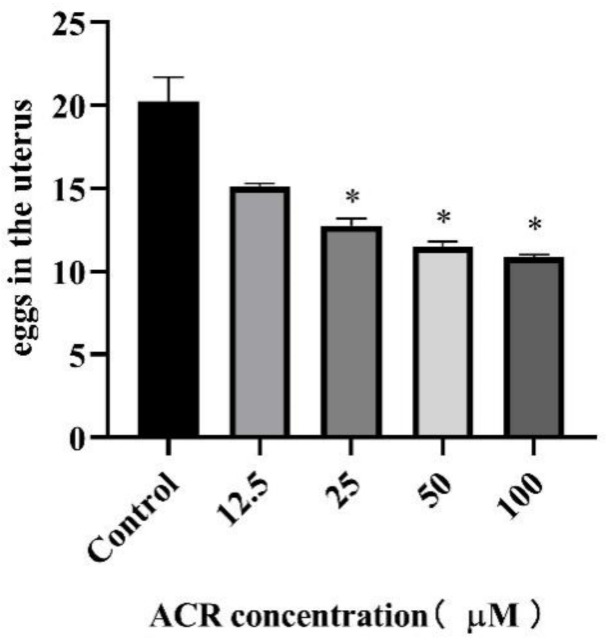
Effect of ACR exposure (12.5, 25, 50, and 100 µM for 48 h) on eggs in the uterus of *C. elegans*. Values are means ± SD. * Compared with the control group, *p* < 0.05.

**Figure 5 foods-11-01590-f005:**
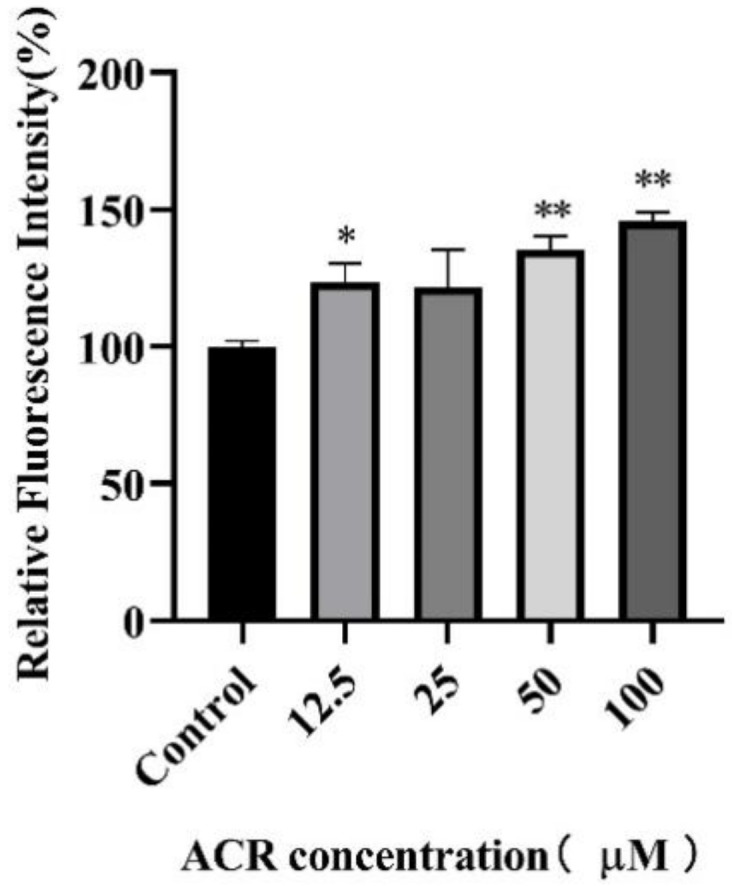
Effect of ACR exposure (12.5, 25, 50, and 100 µM for 48 h) on the ROS activity of *C. elegans*. Values are means ± SD. * Compared with the control group, *p* < 0.05; ** compared with the control group *p* < 0.01.

**Figure 6 foods-11-01590-f006:**
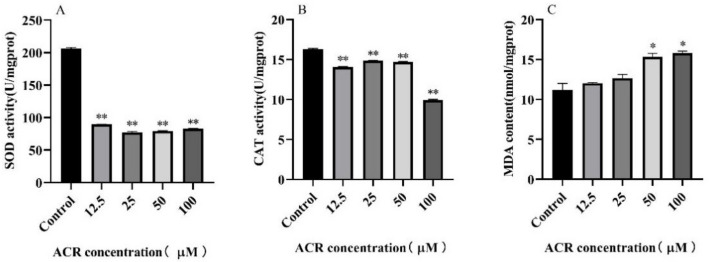
Effect of ACR exposure (12.5, 25, 50, and 100 µM for 48 h) on the antioxidant enzyme activities, namely (**A**) SOD activity, (**B**) CAT activity, and malondialdehyde content (**C**) MDA content, of *C. elegans*. Values are means ± SD. * Compared with the control group, *p* <  0.05; ** compared with the control group, *p* < 0.01.

**Figure 7 foods-11-01590-f007:**
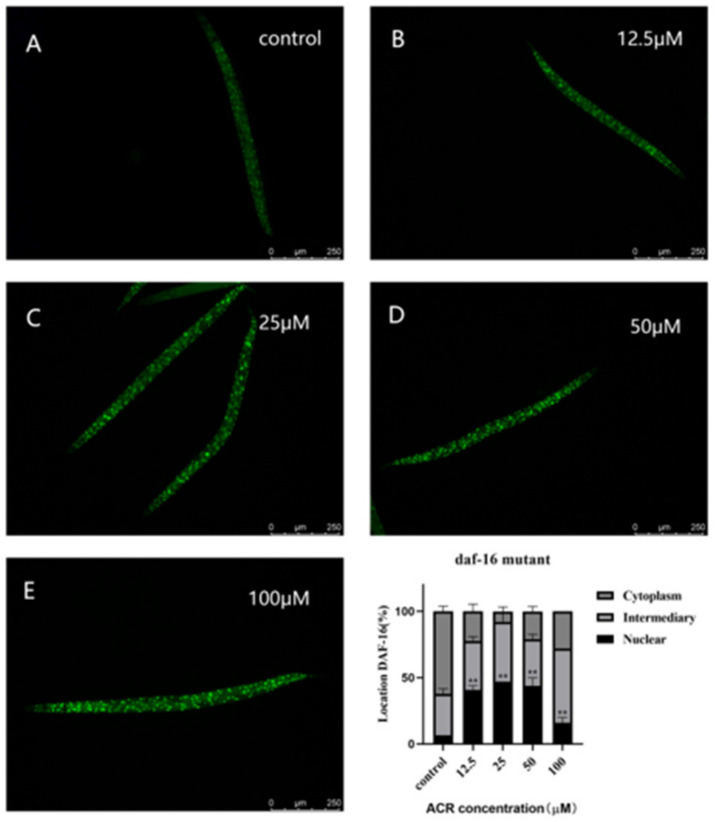
Effect of TJ356 stain exposed to four concentrations (12.5, 25, 50, and 100 µM) of ACR on subcellular DAF-16 nuclear localization (*n* ≥ 30 worms per group). Cytoplasm: The green fluorescence was diffusely distributed in the cytoplasm (**A**). Nuclear: The green fluorescence was concentrated in the nucleus and presented granularity (**C**–**E**). Intermediary: The green fluorescence was between the two states (**B**). ** compared with the control group, *p* < 0.01.

**Figure 8 foods-11-01590-f008:**
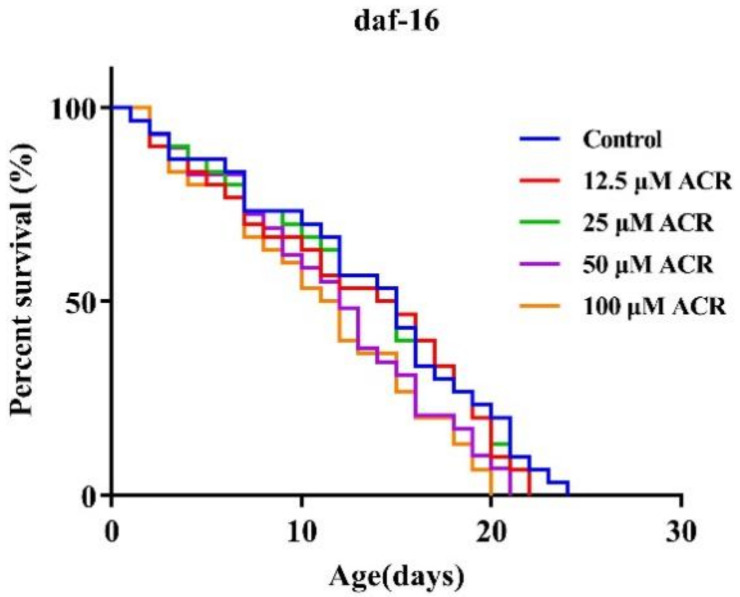
Effect of ACR exposure (12.5, 25, 50, and 100 µM for 48 h) on lifespan of *daf-16* mutant (*n* ≥ 30 worms per group).

**Figure 9 foods-11-01590-f009:**
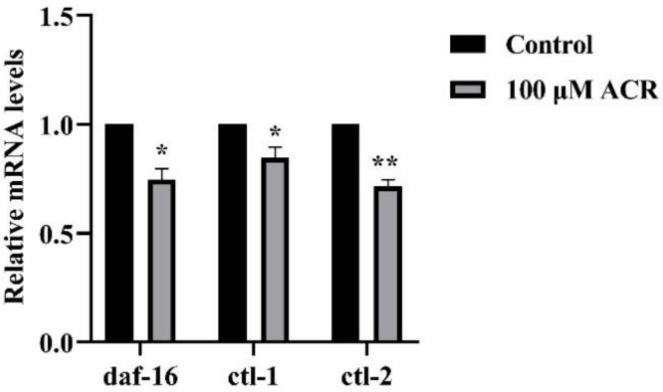
Effect of ACR exposure (100 µM for 48 h) on the relative mRNA levels of *C. elegans*. Values are means ± SD. * Compared with the control group, *p* < 0.05; ** compared with the control group *p* < 0.01.

**Table 1 foods-11-01590-t001:** Effect of ACR exposure on the lifespan in of N2 nematodes.

Group	Dose (μM)	Average Lifespan
Control	0	16.53 ± 0.94 ^a^
ACR	12.5	14.83 ± 0.27 ^a^
	25	14.32 ± 0.32 ^b^
	50	13.32 ± 0.22 ^b^
	100	9.97 ± 0.70 ^c^

Values represent mean ± standard deviations. Significant differences (*p* < 0.05) in the same column were expressed by using different letters (^a^, ^b^, ^c^).

**Table 2 foods-11-01590-t002:** Effect of ACR exposure on the activity of antioxidant enzymes and malondialdehyde content in *C. elegans*.

Group	Dose (μM)	SOD	CAT	MDA
Control	0	206.20 ± 1.56 ^a^	16.31 ± 0.08 ^a^	11.18 ± 0.82 ^c^
ACR	12.5	89.46 ± 0.177 ^b^	14.05 ± 0.08 ^b^	12.03 ± 0.09 ^bc^
	25	77.35 ± 1.47 ^c^	14.87 ± 0.03 ^c^	12.66 ± 0.46 ^b^
	50	79.56 ± 0.35 ^d^	14.69 ± 0.08 ^d^	15.34 ± 0.41 ^a^
	100	83.17 ± 0.53 ^e^	9.94 ± 0.09 ^e^	15.80 ± 0.26 ^a^

Values represent mean ± standard deviations. Significant differences (*p* < 0.05) in the same column were expressed by using different letters (^a^, ^b c^, and ^d^). SOD, U/mg prot; CAT, U/mg prot; MDA, nmoL/mg prot.

## Data Availability

The data that support the findings of this study are available from the corresponding author upon reasonable request.

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
