# Peer review of "Acrolein Promotes Aging and Oxidative Stress via the Stress Response Factor DAF-16/FOXO in Caenorhabditis elegans"

_foods, 2022, doi:10.3390/foods11111590_

Round 1
Reviewer 1 Report
This is a study dealing with the toxic effects of ACR on C. elegans.
Please consider the points given below
Abstract
Line 13: “development rate”. I think the word rate should change. Or you may only leave was is written in the parenthesis “body length and body width”.
Line 18: “diminished of lifespan shortening trend”. Please rephrase
At this section you imply that only the concentrations of 50 and 100 μM had an effect on C. elegans. If somebody reads only the abstract understands that at lower concentrations there was no observed effect, something that is opposite to your results, as presented through the manuscript. Or you should add the respective concentration(s), or you may add a phrase to clarify, such as: major effect/mostly affected….
Introduction
Lines 24-25: it is attention not attentions
Line 35: “under a significant priority poisonous environmental pollutant”. Please correct the sentence. Syntax errors
Lines 41-42: “It is not complicated to cultivate worms in the laboratory, since their ability to reproduce in adults in a short period of time and their short physiological cycle.”. Please rephrase. I think something is missing so as to make the sentence understandable.
Line 44: “and majority human disease pathways are available”. Please rephrase
Line 49: disease or diseases? Give also a reference(s).
Lines 49-50: “However, information on ACR toxicity to C. elegans is not available, and their effects on the oxidative stress and aging from the DAF-16/FOXO pathway are limited.” If the meaning is retained, please write as follows: “However, information of ACR toxicity on C. elegans is limited, as well its effect on the oxidative stress and aging process related to the DAF-16/FOXO pathway”
Line 55: that participate instead of participated
Line 61: ROS equals to Reactive Oxygen Species
Results and discussion
Line 137: add the word animals after wild-type
Line 138: treated instead of treatment
Line 141: indicates instead of found
Lines 142-143: “suggesting that ACR had a sustained negative effect on the lifespan of C. elegans.”. I suggest to delete this sentence. It is a repetition of what you write above (lines 141-142). In addition, you repeat the same at lines 145-147
Line 147: “with the concentration of ACR”. I suggest to delete this sentence
Line 165: “given that” instead of given
Line 185: “as presented at figure 3” instead of “As Fig. 3 showed”
Line 213: “which is consistent” instead of “which was consistent”
Line 219: “is attributed” instead of “be attributed”
Line 222-223: “therefore, ROS production is a common indicator of oxidative stress”. This sentence is unneeded. Same at line 233: “represented by strong ROS levels”
Lines 244-245: please rephrase the following sentence “, CAT enzyme activity decreased from 16.31±0.08 Umol/mg prot (0 µM) to 9.94±0.09 Umol/mg prot (100 µM), and decreased by 39.06%”
Line 261: cancel the word relationship
Line 263: replace “retardation of behavior” with “behavioral retardation”
Line 292: “Previous results”. Previous results of which study? Please rephrase or add reference
Lines 301-302:” indicating that the inhibitory effect of ACR on lifespan was reduced significantly as daf-16 knockout”. I think this sentence is unneeded. Otherwise please rephrase without writing a repetition of lines 209-300
Lines 305-309: the sentence is too big.
Line 307: please rephrase the following: transforming DAF-16 in cytoplasm into nucleus
Figure 7: the legend is not correct
Figure 1: please check again if the asterisk (*) or (**), is explained correctly. In addition, check all figures legends as sometimes asterisk (*) or asterisk (**), is not appeared at the diagrams
Figure 9: The legend refers to all the concentrations used. The diagram depicts only one concentration, that is 100 μM. According to the text (section 3.7), only one concentration was used to study the expression of ctl-1 and ctl-2 after exposure to ACR. Please justify why you choose the highest concentration, since according to your results also treatment with minor concentrations had a considerable effect for example on E. elegans lifespan e.t.c.
Throughout the text/images/tables please check if the word “effects” is correctly used or if it should be replaced with the word “effect”
Throughout the text, please check if the word speculate/speculated is properly used
Throughout the text, please check if figures are placed correctly. For example, figure 3.
- I do understand and appreciate that the work done from the authors requires great effort. However, I think I do not understand the novelty of the work done. Although clinical trials and in vivo experiments are limited, acrolein is in general a very well-studied compound (5.048 results at Pub-Med). Its toxicological profile has been established and limits of exposure have been determined by official health organizations, for example the FDA. I think you should give much more emphasis to the novelty of your work, not by explaining the advantages of C. legans as an experimental model, but by justifying the reasons of yours too many experiments and how your work contributes to the existing knowledge regarding ACR toxicity. Make clear what is the new knowledge you provide.
- Please explain the following: according to which criteria you choose the concentrations tested and please try to contrast the concentrations used with that established by WHO (7.5 μg/kg).
Other comments
At the section “Results and Discussion” Discussion is quite inexistent. Please improve by giving emphasis in explaining better your results and not only mention/summarize them.
In general, the work done needs some improvement. First of all, English language should be very carefully checked. Some points have been corrected. But still many points need to be improved. Authors should be careful regarding the correct use of past and present tense as well as the correct use of some more “complicated” words. Maybe simple words match better.
Thank you for your consideration
Reviewer 2 Report
The article named “Acrolein promotes aging and oxidative stress via the stress response factor DAF-16/FOXO in C. elegans” contains a logical methodology to reach the objectives of the study. It was careful taking into account at first the more “obvious” differences in C. elegans as life span, development, locomotive behavior and reproduction in specific doses of acrolein (0, 12.5, 25, 50 and 100 μM).
Moreover, investigating further the underlying reasons reaching to the conclusion that the ROS levels were increased due to the lower transcriptional activity of DAF-16/FOXO in presence of higher concentrations of acrolein.
Few revisions need to be made in this article:
- At line 103 there is a typo. It should state: “as mentioned before”
- For figure 1B they should remove the label “median lethal time” from the X axis and readjust the graph labels. For example as this:
Y axis should be labelled as LT50 (Days), which is the typical way to define the medial lethal time. X axis is the ACR concentration (μM).
Reviewer 3 Report
The manuscript is an interesting work about the toxicity of acrolein in C. elegans. The authors could demonstrate that acrolein induces oxidative stress, aging and disturb the reproduction of this roundworm. The document is well prepared and is scientifically sound. There are, however, some points that need revision.
Line 192 and 205: Fig 3 and fig 4 need to be switch. They are in the wrong section
Line 220: wrong section name
Line 245 and 248: The units for the enzyme’s activity (Umol/mg) should be corrected. You can use U/mg, but you also need to define what corresponds to 1 U of activity, eventually in M&M section. e.g. decompose 1 uM of XX per min at pH ?? Umol/mg does not make sense.
Line 256: You mention biofilms in the sentence. In what sense?
Figure 6 and table 2: They are 2 different layouts of the same data. However, the data from MDA quantification is quite different. In the figure, it ranges ≈ between 10-15, and in the table between 1.7-2.9. Why?
